# Ongoing training of community health workers in low-income and middle-income countries: a systematic scoping review of the literature

James O'Donovan,[1] Charles O'Donovan,[2] Isla Kuhn,[3] Sonia Ehrlich Sachs,[4] Niall Winters[1]

[1]Department of Education, University of Oxford, Oxford, UK
[2]Faculty of Medicine and Health, University of Leeds, Leeds, UK
[3]Medical Library, School of Clinical Medicine, University of Cambridge, Cambridge, UK
[4]Earth Institute, Columbia University, New York City, New York, USA

**Correspondence to**
Dr James O'Donovan;
james.odonovan@seh.ox.ac.uk

## ABSTRACT

**Objectives** Understanding the current landscape of ongoing training for community health workers (CHWs) in low-income and middle-income countries (LMICs) is important both for organisations responsible for their training, as well as researchers and policy makers. This scoping review explores this under-researched area by mapping the current delivery implementation and evaluation of ongoing training provision for CHWs in LMICs.

**Design** Systematic scoping review.

**Data sources** MEDLINE, Embase, AMED, Global Health, Web of Science, Scopus, ASSIA, LILACS, BEI and ERIC.

**Study selection** Original studies focusing on the provision of ongoing training for CHWs working in a country defined as low income and middle income according to World Bank Group 2012 classification of economies.

**Results** The scoping review found 35 original studies that met the inclusion criteria. Ongoing training activities for CHWs were described as supervision (n=19), inservice or refresher training (n=13) or a mixture of both (n=3). Although the majority of studies emphasised the importance of providing ongoing training, several studies reported no impact of ongoing training on performance indicators. The majority of ongoing training was delivered inperson; however, four studies reported the use of mobile technologies to support training delivery. The outcomes from ongoing training activities were measured and reported in different ways, including changes in behaviour, attitudes and practice measured in a quantitative manner (n=16), knowledge and skills (n=6), qualitative assessments (n=5) or a mixed methods approach combining one of the aforementioned modalities (n=8).

**Conclusions** This scoping review highlights the diverse range of ongoing training for CHWs in LMICs. Given the expansion of CHW programmes globally, more attention should be given to the design, delivery, monitoring and sustainability of ongoing training from a health systems strengthening perspective.

## INTRODUCTION

The WHO have forecast a global shortage of 18 million health workers by 2030.[1] One solution to address this gap has been to advocate for the recruitment, training and deployment of community health workers (CHWs) in low-income and middle-income countries (LMICs).[2] In the broadest sense, CHW is an umbrella term for lay people working within their own community in a health promotion, prevention and delivery role[3]; however, the nomenclature used to describe CHWs is wide ranging and their exact roles, responsibilities, recruitment, remuneration and training vary from country to country.[4 5] When provided with the correct resources, training and support, CHWs have been proven to help improve health outcomes and accessibility to basic services.[2 6 7]

The WHO have suggested that for CHWs to fulfil their role successfully, they require 'regular training and supervision'.[8] For the purpose of this scoping review, we will focus specifically on evaluating the provision of ongoing training for CHWs, rather than initial or preservice training, since ongoing training has typically been 'the most neglected phase' of training,[9] with significant variability in terms of how it is delivered.[10]

### Strengths and limitations of this study

► The study searched 10 major databases and additional non-peer-reviewed literature in order to include as many relevant studies as possible.
► This is one of the first known reviews to assess the provision of ongoing training to community health workers (CHWs) in low-income and middle-income countries.
► Methodological quality assessment of the studies included did not take place, since this was a scoping review.
► Due to the exclusion criteria and search strategy, some non-governmental organisation (NGO) or intergovernmental organisation (IGO) reports may not have been included in this review.
► There is no fixed definition of a CHW, so some exclusions based on terminology may be debated.

Ongoing training includes 'in-service' or 'refresher' training, defined as 'follow on training received after a period of initial training',[11] or supportive supervision, defined as 'a process of helping staff to improve their own work performance continuously… with a focus on using supervisory visits as an opportunity to improve knowledge and skills'.[12]

Despite the importance placed on ongoing training,[10] there is significant variation both in terms of its frequency, content, structure and monitoring between the different groups responsible for training CHWs.[10 13–15] For example, a study by Singh et al[13] found there were 22 different designated organisations responsible for training CHWs in Uganda. The study also found that many of these organisations did not have specific training on 'when, what and how to supervise' CHWs.[13]

The frequency with which ongoing training is provided appears to vary significantly between different organisations and countries. Guidelines produced by the The United States Agency for International Development (USAID) Health Care Improvement Project recommend that refresher training should be provided at least every 6 months to update CHWs on new skills, reinforce initial training and ensure they are practising skills learnt,[16] yet some CHWs have not had refresher training for over 5 years.[17] This finding of a poor provision of ongoing training is commonplace and mentioned in several other studies, across multiple geographic contexts.[18–21] A multinational analysis from several countries in sub-Saharan Africa concluded that the current provision of refresher training courses was 'not sufficient to meaningfully improve the quality of care in these countries', raising into question the need to assess the effectiveness of training programmes, both from the perspective of the individual CHW and the health system in which they operate.[22]

Although a systematic review was published in 2013 by Bluestone et al[23] evaluating effective inservice training design and delivery for health professionals more broadly, there has been no review to specifically assess ongoing training for CHWs in LMICs. A review published in 2014 by Hill et al[24] aimed to determine the impact of supportive supervision strategies for health workers in LMICs; however, the scope of this review was relatively narrow, focusing just on supportive supervision, rather than ongoing training more broadly and included multiple cadres of health workers.

The aim of this systematic scoping review was therefore to map the current delivery, implementation and evaluation of ongoing training provision for CHWs in LMICs.

## METHODS
### Review approach
We conducted a systematic scoping review on the provision of ongoing training for CHWs in LMICs. A scoping review is defined as 'a form of knowledge synthesis that addresses an exploratory research question aimed at mapping key concepts, types of evidence, and gaps in research related to a defined area or field by systematically searching, selecting, and synthesizing existing knowledge'.[25] Scoping reviews are part of the family of research synthesis methods, but compared with systematic reviews address broader research questions. They aim to provide an overview and organisation of existing knowledge rather than a narrow synthesis of a predefined research question[26 27] and place less emphasis on the critical appraisal of the included evidence compared with a traditional systematic review.[28]

A scoping literature review was chosen for this study since we wished to discover the gaps in the literature with regards to the provision of ongoing training for CHWs in LMICs—an area that has not been reviewed before. This approach also enabled us to review a broad body of literature to better understand the current landscape of ongoing training across a variety of contexts. This included mapping the extent, range and nature of how ongoing training is provided and what future research needs to be undertaken.

A review protocol was not published, and the study was not registered with The International Prospective Register of Systematic Reviews (PROSPERO), as these mechanisms are not applied to scoping reviews.[25 26] Nonetheless, our scoping review followed explicit and transparent research steps to explore the research evidence on ongoing training for CHWs in LMICs.

### Search strategy and selection criteria
The Cochrane Library, The Campbell Collaboration and PROSPERO and grey literature were searched to identify available or ongoing systematic reviews pertaining to the provision of ongoing training for CHWs in LMICs. No previous or ongoing relevant reviews were identified.

We then designed an exhaustive and sensitive search strategy to identify all relevant studies. The search was developed with and reviewed by a medical librarian (IK) to ensure completeness. The search strategy was deliberately designed to be over inclusive. Thirty-seven relevant search terms for 'Community Health Workers' and 'on-going training' were developed (see online supplementary table 1 *for the full list of terms used within the search strategies*). These were combined with the World Bank Group 2012 list of LMICs[29] using the AND boolean operator to develop a master search string. Where appropriate, each index-linked Medical Subject Headings term was exploded to contain all relevant subheadings. In addition, synonyms were searched for each key term, along with wildcards and truncation for free-text words. A full record of the conducted search for each database is provided in the online supplementary material. The following databases were searched to identify primary, peer-reviewed studies published from 12 September 1978, up to and including 10 July 2017:
► MEDLINE.
► Embase and The Allied and Complementary Medicine Database (AMED) via Ovid.
► Global Health via Ebsco.

- Web of Science.
- Scopus.
- Applied Social Sciences Index and Abstract (ASSIA) via ProQuest.
- Literatura Latino Americana em Ciências da Saúde (LILACS).
- British Education Index.
- Education Resources Information Center (ERIC).

We wanted to ensure coverage of the relevant literature and education and the social sciences as well as medical sciences, hence including ERIC, BEI, ASSIA and Web of Science. We also wanted to ensure broader coverage of global literature, hence the inclusion of LILACS, which gives extensive coverage of Latin America and the Caribbean. The 12 September 1978 was chosen as a cut-off date, since this was the date of the Alma Ata Declaration, which identified CHWs as 'one of the cornerstones of comprehensive primary health care'.[8]

Despite issues relating to data quality, we included non-peer-reviewed literature in this review in order to encapsulate a broad overview of the literature pertaining to refresher training for CHWs in LMICs. To identify relevant additional non-peer-reviewed literature, we used the following sources: e-theses online service, conference proceedings on Index of Conference proceedings and Google Scholar. Finally, we also searched the reference lists of all relevant papers that we identified, using snowball sampling.

### Inclusion and exclusion criteria

Studies were included if:
1. The primary participants were CHWs.
2. The CHWs worked in a country defined as low income or middle income according to World Bank Group 2012 classification of economies.
3. It was explicitly stated that the objectives or aims of the study were to evaluate or assess the provision of ongoing training, which could include refresher training, inservice training, continuing training or supportive supervision.

Studies were excluded if:
1. The primary focus of the paper was on healthcare professionals other than CHWs; for example, doctors, medical students, nurses or allied healthcare professionals, such as midwives or community-based physician's assistants, were excluded.
2. The study was not conducted in a country defined as a LMIC according to World Bank Group 2012 classification of economies.
3. The paper was not an original, full text, research study. For example, commentaries, letters, opinion pieces, study protocols, training needs assessments and conference proceedings with only an abstract available, were all excluded.
4. The focus of the study was primarily on initial or preservice training, rather than on-going training.
5. As part of the screening process during the full-text review stage, studies were excluded if they did not report or describe the following three areas: (1) the design, (2) the duration and frequency and (3) the outcomes of the ongoing training programme. It was deemed necessary that these three areas were commented on in order that we had sufficient detail about the ongoing training programme from which to base our analysis. These were also good screening questions from which to exclude studies for which the description and evaluation of ongoing training was not the primary focus of the study but rather was just mentioned briefly or in passing.

Since the aim of our scoping review was to map the existing literature regarding the provision, design and outcomes of ongoing training, both qualitative and quantitative study designs were included. Studies did not require a comparison group for inclusion.

### Population

Although the nomenclature given to CHWs varies across the literature, for the purpose of this study, we referred to the 2007 WHO definition:

Community health workers should be members of the communities where they work, should be selected by the communities, should be answerable to the communities for their activities, should be supported by the health system but not necessarily a part of its organization, and have shorter training than professional workers.[30]

This definition allows for different types of healthcare workers to be classified as CHWs in different contexts. To clarify the ambiguity surrounding the term 'shorter training' given in the description above, we followed the definition from Lewin et al,[31] to define shorter training as: 'no formal professional or paraprofessional certificated or degreed tertiary education'.

### Intervention

Studies had to focus on the provision of ongoing training. For this review, ongoing training is an umbrella term referring to any type of training a CHW can receive after a period of initial training. This can include refresher training, continuing training, inservice training or supportive supervision. We purposely aimed to encapsulate a broad range of ongoing training subtypes, so as to better understand the current state of the field.

### Research design

To be included, studies had to qualify as an original, full-text, research study. This meant that review articles, commentaries, letters, policy briefs, protocols, training needs assessments and conference abstracts were not included. Generally, the original article had to include an introduction, explicitly state that the aim of the study was to evaluate the provision of ongoing training and include a methods, results and discussion section to allow us to extract the necessary data for the questions we set out to answer.

## Outcomes

No studies were excluded based on the measured outcomes, since one of the primary aims of this scoping review was to determine which measures are used to report the outcomes of ongoing training programmes.

## Study selection

All papers identified via database searching were exported into EndNote 7.1, and duplicate references were removed. Titles and abstracts of all publications identified in the search were screened by two authors (JO and CO). This determined whether they would be considered for a full-text review. Those that were clearly irrelevant to the topic of this study were discarded at this stage. The full text of all the papers identified as potentially relevant by one or both review authors was then retrieved and reviewed in full against the inclusion and exclusion criteria. At all stages, disagreements between the review authors were resolved via discussion or, if required, by seeking a third review from an independent researcher. The independent researcher was always the same person and was not part of the direct research team listed in this study. Where appropriate, we contacted the authors of individual studies for further information.

## Data analysis

Once studies were determined to have met the inclusion criteria, the relevant data was systematically extracted from each study and tabulated using a 'data charting form' in a Microsoft Excel spreadsheet by one author (JO). The data extracted from each study included the study author, title, date, country and region in which the study took place, CHW name and cadre description, the number of CHWs who took part in the study, the disease focus area, a description of how the ongoing training programme was delivered, as well as a report on the outcomes measured. The use of a 'data charting form' has been recommended by Arksey and O'Malley and Levac et al, as a key stage of conducting a scoping review.[26 32] Where necessary, the corresponding authors for relevant studies were contacted via email to clarify aspects of their work prior to final inclusion.

Once the data had been transferred into the spreadsheet, two authors (JO and NW) reviewed the information and selected key focus areas for the review, as well as categories for the outcome reporting methods. The same two authors thematically grouped outcome data from ongoing training into one of the following four categories: (1) knowledge and skills assessments; (2) changes in behaviour, attitudes or practice; (3) qualitative assessments; and (4) mixed methods approaches. Similarly, if the use of mobile technologies was noted in the study, this was documented and categorised using the mHealth framework developed by Labrique et al.[33]

## Patient and public involvement

Patients and the public were not involved in this scoping review.

## RESULTS

### Search results

The initial search of the 10 databases yielded 3923 articles (see online supplementary table 2). After exclusion of duplicate references using the EndNote referencing system, 2609 papers were identified for initial screening. After the initial abstract and title screen, 172 studies were identified for full-text review. Following this review, 137 papers were excluded. Reasons for exclusion at full-text screening can be found in the Preferred Reporting Items for Systematic Reviews and Meta-Analyses flow chart (figure 1). As a result, we were left with 35 original studies meeting the inclusion criteria.[13–15 34–65]

### CHW cadres and study characteristics

Twenty-two different terms were identified as defining CHWs across the 35 studies, with significant variations being noted between studies in terms of CHW roles, responsibilities and status. The majority of studies evaluating the provision of ongoing training for CHWs have been published since 2015 (n=19), with no relevant studies published before 1993. In terms of geographic location, most studies took place in East Africa, (n=16) or South Asia (n=7). For full details regarding CHW cadre descriptions and study characteristics, please refer to online supplementary table 3.

### Ongoing training details

The reported type, frequency, duration, training focus and outcomes of ongoing training delivery were highly variable between studies (see online supplementary table 3). For example, Zeitz et al[65] reported on a 1-day refresher training course for CHWs that specifically focused on acute respiratory illness in children and used pretesting and post-testing of knowledge as the outcome measure of the training. This is in contrast with Kawasaki et al[48], who carried out a 2-year study where CHWs received monthly refresher trainings, and the outcome measures were focused on behaviour change at the community level, for example, improved handwashing techniques and the number of household visits carried out by CHWs. This variation, both in terms of duration, structure and content focus, makes direct comparison between studies difficult (see online supplementary table 3).

Oliver et al,[66] highlighted the importance of codesigning programmes with CHWs to help ensure relevance to their practices and experiences. This scoping review revealed a lack of participatory input from key stakeholders in the design and delivery of the training programmes. Only five studies documented seeking input from CHWs in the design of the training programmes.[13 14 48 56 57] For example, Puchalski Ritchie et al stated that the training content was developed based on the training needs identified by CHWs in a qualitative survey prior to the programme being established.[56 57] They also mentioned that the training sessions and tools were chosen in consultation with local collaborators, and the local language was used in the study.[44 64]

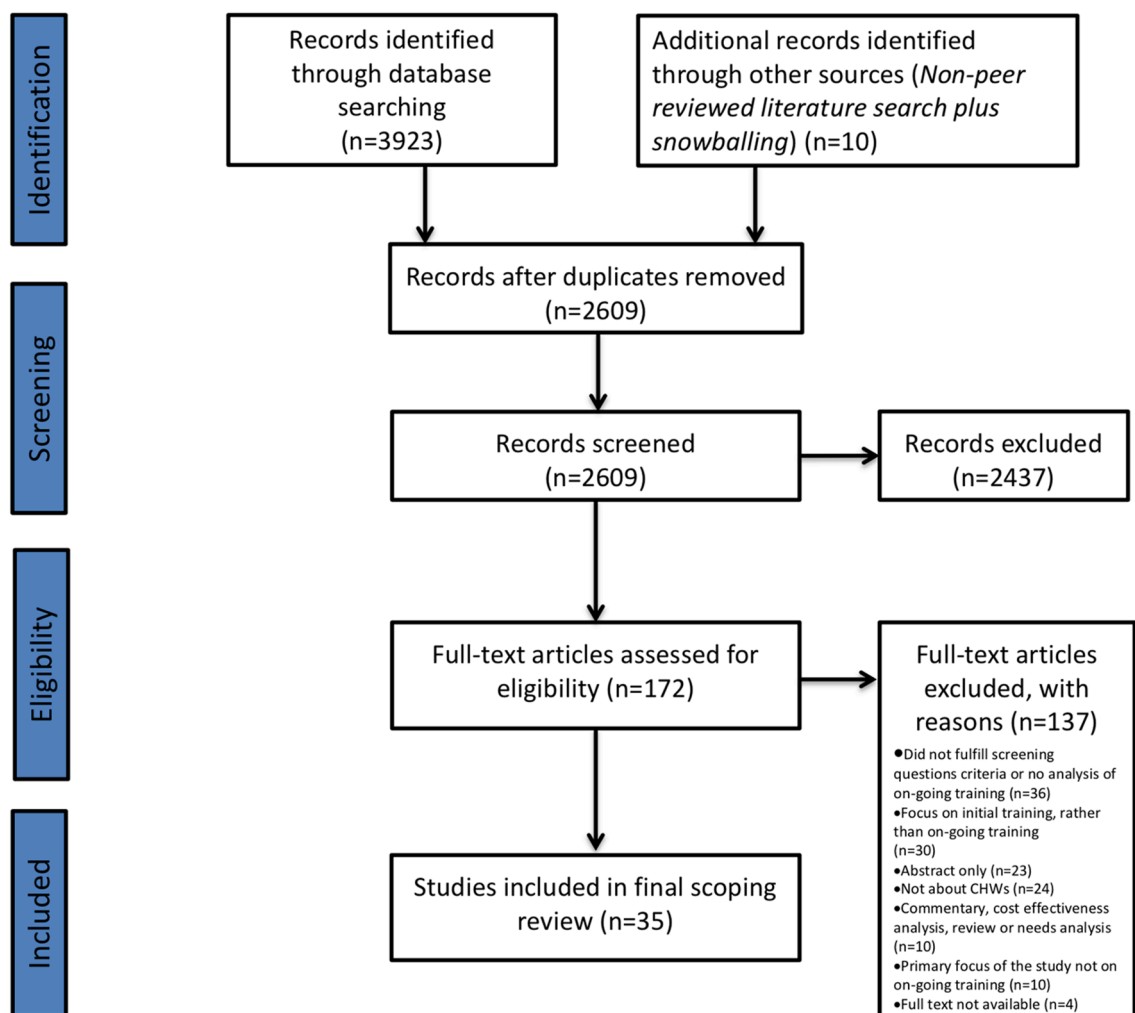

**Figure 1** PRISMA flow diagram. The PRIMSA diagram details our search and selection process applied during the scoping review. CHWs, community health workers; PRISMA, Preferred Reporting Items for Systematic Reviews and Meta-Analyses.

With regards to programme delivery, most training was delivered inperson, with four studies reporting the use of mobile technologies to deliver or assist ongoing training activities.[14 47 59 64] To report the outcomes of ongoing training programmes, a range of different measures were used. The majority of studies evaluated the effect of ongoing training by using proxy markers to assess change in practice, attitudes or behaviour (n=16).[13 34–36 38 47–49 52–54 56 61–64] This included assessing behaviour change at a community level, such as improved vaccination uptake and hand washing among households,[13] to changes in practice among CHWs, such as improved record keeping.[47] Assessment of knowledge and skills, mainly through the use of preintervention and postintervention tests, formed the sole means of evaluation in six of the studies.[40–44 65] Five studies used a qualitative approach for programme evaluation, mainly through the use of interviews and focus discussion groups, which were then thematically analysed and reported.[14 39 46 57 59] Eight studies adopted a mixed methods approach, using a combination of knowledge and skills assessments, qualitative approaches or changes in behaviour, attitudes

and practice.[15 37 45 50 51 55 58 60] The outcomes reported were variable given the heterogeneity of the approaches to evaluation; however, the majority of studies reported positive outcomes following ongoing training; for example, Horwood *et al*[45] found that children managed by a CHW who had attended a refresher training session were more likely to be managed correctly according to iCCM guidelines compared with those who had not. Similarly, a study by Singh *et al*[13] found that homes in areas where CHWs had received supportive supervision were more likely to have installed and functioning tippy taps for hand washing, compared with areas served by CHWs who had not received supervision. Yet, despite the many positive outcomes associated with ongoing training, there were also studies that found no difference in outcome measures between CHWs who received ongoing training and those who had not, and there were even negative reports of ongoing training. One such example of this was the study by Javanparast *et al*[46] that revealed that CHWs were dissatisfied with ongoing training in its current format, in particular 'its quality and timing, the infrequency of courses, inadequately qualified trainers who

are unfamiliar with the behvarz (CHWs) working environment, the lack of practical sessions and of physical space and training facilities'. Similarly, Ndima et al[15] found that CHWs in Mozambique felt their supportive supervision was poorly organised, causing them to feel demotivated, with their supervisors citing high concurrent workloads and a lack of support.

For full details of the outcomes for individual studies, please refer to online supplementary table 3.

## DISCUSSION

There is a diverse range of approaches in the design, delivery and reported outcomes of ongoing training for CHWs in LMICs, and a number of significant gaps remain.

### Location, content and duration of ongoing training programmes

The majority of studies describing ongoing training for CHWs have a narrow geographic concentration. Given the highly contextualised role of the CHW,[67] this presents an opportunity for further research to be carried out in other geographical contexts. Furthermore, the majority of studies focused on the provision of ongoing training for maternal and child health or infectious diseases such as HIV and tuberculosis. Given the combined shortage of a lack of specialist health workers and the high morbidity and mortality from the aforementioned disease groups, CHWs have rightly been trained to address these issues. Although the burden of infectious disease and child and maternal health remain problematic in LMIC settings, no studies focused on the provision of ongoing training for non-communicable diseases (NCDs) and only one study focused on the provision of ongoing training for CHWs involved in mental healthcare.[51] NCDs have been described as the 'social justice issue of our time',[68 69] since they disproportionally affect populations in LMICs.[70] It is therefore imperative that more attention is directed towards providing ongoing training in the prevention and management of NCDs at a community level if we are to make realistic progress towards SDG 3.4, which has set the target of reducing premature mortality from NCDs by a third, by 2030.[71] This public health need to expand CHW provision towards NCDs is both an opportunity and challenge, since it will require the commitment of governments, funders and programme managers to retrain and refocus large CHW workforces.

### Delivery of ongoing training programmes

The majority of ongoing training was delivered in person, with only four studies reporting the use of mobile technologies as playing a role in training delivery. This was a surprising finding since mobile technologies have been used as a mean to train other cadres of healthcare professionals in LMICs.[72–74] Given the high ownership of mobile phones in sub-Saharan Africa,[75] and the ability for flexible learning, data collection,[76] the use of mHealth to facilitate ongoing training warrants exploration.[77] One

of the studies included in this review highlighted the role of mobile phones to strengthen supportive supervision for CHWs in Kenya.[14] A WhatsApp group to facilitate instant messaging was created for CHWs and their supervisors to 'support supervision, professional development, and team building'.[14] Importantly, the authors of this study reported on the quality assurance and information exchange, which the system facilitated, and on the supportive environment fostered by the use of the technology.[14]

Given that several studies cited supervisors' high concurrent workloads as to why ongoing training was poorly organised and delivered,[15 39 46 60] mHealth should be explored further as a potential tool to manage human resource shortages, since this is one of the key applications of mHealth tools mentioned by Labrique et al,[33] as a health systems strengthening innovation.

As a caveat, Hampshire et al[78] have urged researchers and practitioners to proceed with caution and consider the financial implications when considering mobile technologies as a training tool for CHWs, due to the potential risk of reinforcing socioeconomic, geographical and gender inequalities. Furthermore, Joos et al,[47] highlighted the need to consider how mobile phones can successfully transition to scale following pilot studies.

### Outcome measures and outcomes of ongoing training

Given the variation of how ongoing training programme outcomes were evaluated and reported, direct comparison between studies is difficult.

For outcome reporting, 16 studies used markers of behaviour change at the household level or CHW practice to measure the impact of ongoing training. Using measures of behaviour change to evaluate the effectiveness of ongoing training is a welcome move towards ensuring meaningful programme evaulation[79 80]; however researchers and programme managers should be aware of the multiple confounding variables that could influence these behaviours, such as the Hawthorne effect, and the difficulty in assessing these practices longitudinally, as well as the need to approach programme evaluations from a complex interventions standpoint.[81]

Similarly, where pretest and post-tests of knowledge and skills acquisition are used to evaluate the impact of ongoing training programmes, they do not necessarily reflect the abilities of CHWs to perform their role well in the community,[82] nor do they provide any insight into CHWs experiences of training. Hamilton and Friesen argue that instrumental views of assessing learning often fail to capture the practical and emancipatory concerns of learners,[83] and thus alternative methods of evaluation should be explored.[84] Furthermore, it is important to consider the validity and applicability of such tests to real-life settings, given that many of the assessment tools have been designed by the researchers and are unvalidated. What is more, some CHWs have only been in formal education to the level of primary or secondary school, and so this form of assessment may introduce

construct-validity bias. Interestingly, Rowe *et al*[61] used a skills and knowledge assessment tool and found no improvement in scores between the groups of CHWs who took part in refresher training and those who did not. They questioned the usefulness of refresher training based on this outcome; however, they failed to acknowledge the other benefits of ongoing training that they did not measure, such as an improved sense of community, motivation and empowerment.[85]

Puchalski Richie *et al*[56] actively avoided using an 'assessment of knowledge and skills', since they were concerned that it might negatively affect participation in training; instead, they carried out a qualitative evaluation of CHWs overall satisfaction with the programme as a measure of training success.[57] Similarly, other studies, which used a qualitative approach to outcomes evaluation, found that CHWs had negative experiences of ongoing training—insights that would not have necessarily be revealed if a purely empirical approach was taken towards programme evaluation.[86] A mixed methods approach towards evaluation may therefore be a useful approach for future studies.

No studies used the framework for outcome-level evaluation of inservice training of healthcare workers produced by O'Malley *et al*[87] in 2013. This framework was developed in a holistic manner to evaluate inservice training of health workers based on the needs of the individual, the organisation and the health system. Current assessment of inservice training programme assessment relies heavily on measuring and reporting training 'outputs' such as the number of CHWs trained, the total hours of training delivered and scores obtained on standardised tests.

A small number of studies used self-reported satisfaction,[64] motivation[58] or increased agency[57] as outcomes to measure the impact of ongoing training. These are what Kok *et al*[88] refer to as 'software' of a training programme and can affect motivation and performance. Kok *et al* argue that the software elements of the system are important since they affect CHW performance by 'influencing self-esteem, attitudes and agency',[88] as well as satisfaction and motivation. Ndima *et al*, commented that when training focuses too heavily on developing technical skills there is a danger that 'examining value and attitudes of CHWs and abilities to understand and support individual and group dynamics'[15] can be lost.

### Participatory approaches to ongoing training design, delivery and evaluation

Given the lack of documented participant input and feedback in terms of programme design, delivery and evaluation, research into the use of participatory action research (PAR) is one area that would warrant further investigation. PAR broadly involves working 'with' end-users in a collaborative effort rather than 'for' or 'on' them.[89] It encapsulates the ideals of promoting autonomy and social justice and works on the principle that the end-users wishes and needs, should be respected and valued.[89] This school of thought was echoed by Perry and Crigler,[90] who advised a

'top-down supervisory approach… may not be as feasible or effective as a participatory supervision model where CHWs and their communities are provided with the resources and autonomy to seek out the support that they need to perform well and stay motivated'.

It is also important to consider sociocultural sensitivities in the design of an ongoing training intervention, including cultural beliefs, especially in areas where the practice of traditional medicine is still commonplace and may be at odds with a more Western approach to healthcare. In the study by Singh *et al*, the training intervention was delayed by 4 months due to villagers believing the immunisations used by the CHWs were intended to cause infertility and the insecticide treated bednets were designed to 'kill their children'.[13] This is especially relevant when ongoing training programmes are being designed and implemented by non-native researchers, in countries emerging from postcolonial pasts and where local beliefs are rooted in historical antecedants.[91]

### Study limitations

It is important to recognise that given the highly contextualised nature of CHW training programmes,[21 67] this scoping review does not try to address best practice or provide guidelines. Rather, we have attempted to map the current landscape of ongoing training for CHWs in order to broadly identify key similarities or differences between ongoing training programmes and identify areas that may have received little attention in the literature to date to help inform other researchers, practitioners or policy makers working in this field.

We tried to be as inclusive as possible to identify relevant literature, but with the diverse range of terms used to describe CHWs, it is possible we have inadvertently missed out some eligible studies describing ongoing training for CHWs. Furthermore, we did not conduct an exhaustive search for grey literature sources due to the challenges in appraising these types of publications as well as the lack of standardised search guidelines for scoping reviews.[92]

Finally, given the nature of scoping reviews, a critical appraisal of the studies included in the review was not performed.[26] This could be perceived as a limitation since the overall quality and level of detail of the studies was variable. There was also significant heterogeneity between studies, which makes direct comparisons difficult. Future work should aim to clearly outline the context in which CHWs work and provide a detailed description of their job roles and responsibilities to help orientate the reader and contextualise the setting.

### CONCLUSIONS

There is significant variability between ongoing training programmes for CHWs in LMICs, both in terms of design, structure, content, duration and reported outcomes. This fragmented approach means little is understood about how to best deliver ongoing training in LMICs. Ongoing training programmes have largely

taken an empirical approach, focusing on specific areas, for example, child and maternal health and infectious diseases, in limited geographic contexts and have variable approaches towards outcome measurement and reporting. The danger is that this approach fails to acknowledge what Kim *et al*[80] refer to as the 'broader systems and conditions affecting global health care delivery'. Given the heterogeneity of the field, we advocate for a realist approach to evaluation for future research, considering training as a complex intervention. This may help those interested in the field to make better sense of its complex nature with a view to understanding what works, for whom and under what conditions. Through taking this approach and considering the contextual requirements, ongoing training programmes are more likely to contribute to a systems level improvement in resource limited settings.

**Acknowledgements** We wish to thank Ms Stephanie Sobek for proofreading the paper prior to final submission.

**Contributors** Activities undertaken by the authors were as follows: establishment of research question(s) and development of search strategy: JO, IK and NW. Background framing: JO and NW. Database search and record screening: JO, CO and IK. Extraction of primary studies from the included reviews: JO and CO. Discussion: JO, CO, NW and SES.

**Funding** JO is a DPhil candidate at The University of Oxford and is supported by a personal expenses and research support grant from the Economic and Social Research Council (ES/P000649/1).

**Competing interests** JO reports grants and personal fees from the Economic and Social Research Council during the conduct of the study.

**Patient consent** Not required.

**Provenance and peer review** Not commissioned; externally peer reviewed.

**Data sharing statement** All data are contained within the main body of the text and in the online supplementary material.

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
