## [Reviewer comments · BMJ Open]

ARTICLE DETAILS

TITLE (PROVISIONAL)	The on-going training of Community Health Workers in low- and middle-income countries: A systematic scoping review of the literature.
AUTHORS	O'Donovan, James; O'Donovan, Charles; Kuhn, Isla; Sachs, Sonia; Winters, Niall

VERSION 1 – REVIEW

REVIEWER	Andrew Lee The University of Sheffield, UK
REVIEW RETURNED	11-Jan-2018

GENERAL COMMENTS	Thank you for the opportunity to review this interesting article. It certainly is in an under-researched and under-reported area of some importance particularly for LMIC countries that have substantial dependencies on CHW for service delivery and coverage. The writing is of a high standard and is organised and presented well. The methodology is well described. There are a few minor issues which I would like the authors to consider. Firstly, the research question is not strongly articulated. What is presented in the text are 4 questions, each of whom could be a standalone question, which may be confusing for the reader as it does not make clear what the purpose of the review is. I presume that the key question is to review how on-going training for CHW is designed and developed, and that the other elements (e.g. learning theory or mobile technology use) are perhaps subquestions? The questions on learning theory and mobile technology use also introduces in my mind some doubt as to whether the review has an a priori bias. Scoping reviews should help uncover and map gaps in the evidence base(that may include use of technology, theory, but also other elements perhaps?), so it seems counter intuitive to pre-identify the gaps in a way. In terms of methodology, there is some slight blurring here in what has been written - it reads like a cross between a scoping review and a narrative review? If it were a scoping review, it could be a bit broader in scope and try to map various domains around CHW training e.g. what is the starting knowledge base/literacy levels in the different studies reported, what are the different training durations, what are the topic
---

areas (maternal and child health), any underpinning learning theory, how assessed, whether training needs assessments were conducted, and consideration of how the training fits the role (i.e. are training needs role dependent) etc...

The limitations section is poorly developed. The exclusion of the French article is a bit concerning especially if the authors consider that it may be of relevance. I would suggest if at all possible that an attempt is made to include this article, so that the review provides an up-to-date and comprehensive picture, unless the authors are satisfied that this article provides no new insights.

As hinted to by the authors, the term CHW is a catch all for a very diverse category of informal health workers. I suspect the authors are au fait with this as the search terms used (synonyms for CHWs) is quite extensive and covers many of the key ones. I think it would be useful to write a bit more in the introductory section to elaborate this diversity particularly for the non-specialist reader.

The other significant limitation is the exclusion of grey literature. In the development context, there is considerably more published in NGO / IGO repositories, much of which has direct relevance to practitioners and local policymakers. I think this nuance needs to be highlighted.

In the discussion section, the authors make a point that NCD is a priority and that maternal and child health, and HIV/TB seem to predominate. I agree. That said I think it would be useful to illustrate the context better - CHWs exist and are often focused on certain disease programmatic areas because they are cheaper to train and roll out than nursing, medical or paramedical staff in LMICs. The public health need now to reorientate CHW areas of work towards NCDs is essential but it is also challenging as it will require retraining and refocusing large CHW workforces. This challenge is not well made in the discussion.

There was also discussion around the absence of learning theories to underpin the on-going training. That said, the absence of evidence does not necessarily equate to evidence of absence. Neither can it be said that the training approaches adopted are inappropriate or ineffectual on the basis of whether there is an underpinning learning theory described? I don't think on the basis of what evidence is given in the results section a strong assertion can be made on this.

In a similar vein, the authors hint that 'protocol-based' training is inferior or somehow inappropriate. In an ideal situation perhaps. That said, as the authors argue context is everything: do protocol based training approaches predominate because the target audience has low literacy or understanding of science, or there is a higher policy goal of quality assurance and governance that is driving the want for standardized CHW responses rather than autonomy of treatment choices?

The findings and conclusions suggest little robust work in this field but also difficulties making comparisons due to the heterogeneity of the articles. Perhaps the authors may want to consider advocating the value of a realist approach in future research in this field as that may make better sense of the varied data, in terms of the differing contexts, generative mechanisms and their relation to outcomes?

	Overall, a decent piece that I think can be strengthened further and provide some useful additions to the evidence base on this topic.
--	--

REVIEWER	Joseph Mumba Zulu University of Zambia, School of Public Health, Zambia
REVIEW RETURNED	24-Jan-2018

GENERAL COMMENTS	Introduction: This is an important review as it focuses on contributing towards addressing the knowledge gap on refresher trainings among community health workers. This information is a vital considering the increased call or emphasis on the need to promote community-based health systems and workforce. Overall comment: There is need to restructure and strengthen the results section in line with the objective of the review. For now, much information has been placed in the discussion section compared to the results section. There is need to have subheadings in the results section in line with the scoping review objectives or questions, which are: How is on-going training for CHWs designed and delivered? Are theories of learning used to inform the design of on-going training? Do mobile technologies have a role in the delivery of on-going training? How are the outcomes of on-going training reported? In addition, to these questions, the authors may consider including information or a paragraph on the key topics that the community health workers were re-trained in and where possible the duration of the training. This information could be part of the heading under design and delivery of CHW re-training. Minor comments:  - Mention/state the total number of articles that were included for review in abstract - Search terms: while the authors state in the results section that about 17 different terms were identified for community health workers during the search process, the authors do not mention these terms. It will be important for the authors to outline these terms in the methods section – and list them under the search terms used (since the community health worker concept is inadequate as a search term for this kind of study) - Please start the duration for the scoping review - There is a statement that the scoping review is advantageous because of its “less emphasis on the critical appraisal of the included evidence compared to a traditional systematic review”, it would nice if the authors could provide more information on the advantages of this design? - I recommend moving the PRISMA FLOW CHART or steps taken in selecting the final articles from the results section to the methods section - There is need to add data analysis section
--

REVIEWER	Brynne Gilmore Trinity College Dublin, Ireland
REVIEW RETURNED	05-Feb-2018

GENERAL COMMENTS	BMJ Open Review: The on-going training of CHWs in LMICs: A systematic scoping review of the literature February 05, 2018
--

Thank you for the opportunity to read this very interesting study and manuscript. As the authors highlighted, training of CHWs is a very important issue that this manuscript works to address. I hope that you find the below comments helpful in your revisions.

Major:

1. References are mixed up and very confusing. The included articles seem to be referenced separately in the supplementary files from those that are in-text citations, but this makes for duplication. For example, when citing included study in text it might be 31, then in the supplementary file it is 10, so I cannot refer to the data table to get more information. Please fix.

2. Pg. 6 – the research questions seem a bit repetitive (1 and 2). Suggest wording as they were in the abstract. Additionally, question 2 the “Are participatory approaches used to engage communities to ensure context-relevant design” – is new (not in abstract), and also does not necessarily fit with theories. Programmes can use theories, without engaging communities. Should this be a separate question, especially as you title them separately in the Discussion?

3. Much more information is needed on the data extraction and analysis of the included studies. How was the data from included studies systematically extracted? More details on this is needed (i.e. was a template used, and how was this developed, how many people extracted etc). Once extracted, how was this analysed?

4. The Results and Discussion are lacking:

a. The results do not specifically answer your research question, it would be beneficial to draw back to the original questions.

b. The results at times are also just superficial reporting – there is not a lot of information given with them. Even in the Supplementary File 2, the reader does not get enough information to really understand the training. I would suggest giving more details to more fully engage with the included studies and the topic of on-going training.

c. Additionally, a lot of your discussion could belong to results, as not a lot of new literature is brought in to discuss results, it is more of a second reporting of the findings. For instance, Pg 16 paragraph starting on line 45 could all be moved to results. As well as pg. 17 paragraph from line 13 to 32.

d. I think a description of the CHW cadre within the programme should be brought into the Table to provide more context

e. Also, articles that did bring in theories of learning and/or participatory approaches and/or mobile technology, should be indicated in the table. Right now all the pieces are somewhere in the article, but this table should bring them all together.

f. The actual on-going training description is missing. I would hope that one could turn to your article to get details on what training was occurring, but as

of now it is quite limited. More can also be brought in on the type of

	on-going (i.e. who conducted it? What specifically was it on (i.e. Some refresher training focus on 1 unit of a CHW package, others all skills). Who were the supervisors? Where did the supervision and/or training occur?) 1. The 'Outcomes' question needs some more thought I think. There is no clear link between the outcomes and the on-going training, just reporting of what was looked at. While I understand it is a scoping review, so you are reporting what is happening, if an article found no impact on outcomes of a training, I feel like this would be important to note within the article. 5. Please double check all of your data extraction – for the Vallieres article, the 'Outcome' you note is "Qualitative Assessment. Self reported measures of work engagement and job satisfaction". This was done through a self-reported survey – which is quantitative. If you read on, it is all very quantitative with scale validation and statistical reporting. 6. Conclusion – second paragraph. A lot of these ideas (partnerships, interdisciplinary approach, contextual needs, systems level improvement etc), are newly introduced. As well as the recommendations for research. These belong more in the discussion, and then can be reiterated in the conclusion. Minor: 2. Switching between low-or middle-income and low-and middle-income (i.e. pg. 2, pg. 3). Please pick one and change throughout for consistency. 3. Pg. 5 – CHWs often do more than health promotion and prevention – i.e. treatment, testing, surveillance... This is true of the CHWs in some of your included papers. 4. Pg. 5 – implying that CHWs are all voluntary (line 13-17, and then comparing to 'salaried health workers' in line 23). This is not true of all CHWs – and some of your included studies have salaried CHWs (i.e. HEWs in Ethiopia). Please clarify. 5. There are lots of quotes within the paper, but none of page numbers of the reference. Please make sure this is OK with BMJ guidelines, and if not add page numbers to all in-text quotes. 6. Please consider reworking you Supplementary material. Right now it is a little difficult to follow. Also, within reporting supplementary material, please keep the order in-line with the methodology and when it is reported in the article (i.e. move the search terms before the search results). 7. Your description of 'grey literature' is not really consistent with what most would think. Especially given your topic area, grey literature would often include NGO reports, policy documents, programme descriptions and evaluations etc. But these were 1) not searched for looking at your grey lit sources and 2) would have been excluded based on exclusion #3 and research design. So I don't think you include 'grey' literature, but included non-peer reviewed research. At very least, I think you need to specify what type of grey literature. 8. Pg. 14, line 28 – it is not Table 1, as there is a Table before
--	--

	this in supplementary. Again, please consider reworking the supplementary file for more clarity. 9. Mobile training in results – what kind of mobile training was used? 10. Figure 1 – what are the ‘other sources’ is that the grey searching plus snowballing? 11. Supplementary material – search results and strategy. Please do not combine databases when reporting the number of hits. Numbers for each separate search should be reported. 12. If your search is sensitive, I am cautious that BEI only have 1 hit? (and ERIC and LILACS seem low too). Do you have any explanation for this? 13. Page 24, line 14 – “settings of poverty and inequality”. I would change this. It was never brought up before, only that your setting was LMIC. Thus, it reads as you are stating all LMICs are settings of poverty and inequality. What about HICs, there are not settings of poverty and inequality within these? So if this is what you are looking at then these settings should be included too. Additional: 1. Pg. 8 – I would reorder the paragraph at line 42, moving the first sentence to after the second sentence. ... “...are not applied to scoping reviews. Nonetheless, our review followed...., as demonstrated in the following sections.” 2. Pg 13 – the ‘third reviewer’ was independent, as in not part of the research team? Was it always the same person, or how did you find them? 3. Pg. 14, line 13 – Sentence “studies were initially...” Is repetitive with methods section. Results should be reporting only, not describing the screening process again. 4. Pg. 14, line 17 – consider rephrasing “in-depth analysis”, as it could seem that at this point data was analysed, as opposed to articles reviewed/examined.
--	--

VERSION 1 – AUTHOR RESPONSE

BMJ Open reviewers comments and responses:

Reviewer: 1

Reviewer Name: Andrew Lee

Institution and Country: The University of Sheffield, UK

Competing Interests: None declared

Reviewer 1 Comment Number	Reviewer comment	Author response
1	Thank you for the opportunity to review this interesting article. It certainly is in an under-researched and under-reported area of some importance particularly for LMIC countries that have substantial dependencies on CHW for service delivery and coverage. The writing is of a high standard and is organised and presented well. The methodology is well described. There are a few minor issues, which I would like the authors to consider.	Thank you for this positive review and for also recognising that this is an important and under-researched area. We also appreciate your thorough comments and have gone through the manuscript to modify it accordingly with these suggestions in mind. All changes to the manuscript have been made using the 'track changes' feature in Microsoft Word and changes to the supplementary material are highlighted in red. To assist you with your review we have responded to each of your comments in turn within this table, with cross-references to the main text or supplementary material where appropriate.
2	Firstly, the research question is not strongly articulated. What is presented in the text are 4 questions, each of whom could be a standalone question, which may be confusing for the reader as it does not make clear what the purpose of the review is. I presume that the key question is to review how on-going training for CHW is designed and developed, and that the other elements (e.g. learning theory or mobile technology use) are perhaps subquestions? The questions on learning theory and mobile technology use also introduces in my mind	Thank you for highlighting this. You are correct that the broad overall aim of this review was to assess how on-going training is currently delivered. As such, we have attempted to map the current available evidence on on-going training to help researchers and practitioners orientate themselves as to what work has currently been done. As stated in the introduction, we were unable to identify a previous review that specifically assessed the provision of on-going training for CHWs which is why we chose to undertake this work. To ensure clarity with regards to the

	some doubt as to whether the review has an a priori bias. Scoping reviews should help uncover and map gaps in the evidence base (that may include use of technology, theory, but also other elements perhaps?), so it seems counter intuitive to pre-identify the gaps in a way.	research question, we now simply state that: “The aim of this systematic scoping review was therefore to map the current delivery, implementation and evaluation of on-going training provision for CHWs in LMICs.” (Page 6) By revising and simplifying the aims of the paper, the gaps have not been pre-identified and the emergent themes are then discussed within the results and discussion sections of the paper. We have provided a comprehensive table in the supplementary material which outlines all of studies included in the final review, as well as information from each study such as the disease focus area, type of on-going training and total number of sessions provided and outcome measures (see Table 3 in the supplementary material).
3	In terms of methodology, there is some slight blurring here in what has been written - it reads like a cross between a scoping review and a narrative review? If it were a scoping review, it could be a bit broader in scope and try to map various domains around CHW training e.g. what is the starting knowledge base/literacy levels in the different studies reported, what are the different training durations, what are the topic areas (maternal and child health), any underpinning learning theory, how assessed, whether training needs assessments were conducted, and consideration	Thank you for raising this good point. We have made the following changes to make our methodological approach, in line with best practice, more explicit. As stated in response to Comment 2, the primary aim of this review was to map the key areas that may be of interest to policy makers and CHW programme managers and researchers. Taking Reviewer 3’s comments into consideration, we have added further columns to the comprehensive table included within the supplementary material including whether or not studies used mobile technologies to deliver on-going training and whether reference was made

	of how the training fits the role (i.e. are training needs role dependent) etc...	to the use of learning theories in program design or evaluation. We have also added additional information regarding the on-going training, such as who delivered it and its total duration and location (see Table 3 in the supplementary material). In terms of the review design we have adhered to the guidelines published by Arksey and O'Malley in 2005, which outline the methodological framework for conducting scoping studies. We have referenced this within the methodology section of the paper (Page 7). (See Arksey, H., & O'Malley, L. (2005). Scoping studies: towards a methodological framework. International Journal of Social Research Methodology, 8(1), 19-32. doi: 10.1080/1364557032000119616) Although we have tried to report on the level of education at baseline for each cadre of CHW, there was a lack of information within some of the included studies about this. Where possible we have included this within Table 3 under the column 'Cadre Description'.
4	The limitations section is poorly developed. The exclusion of the French article is a bit concerning especially if the authors consider that it may be of relevance. I would suggest if at all possible that an attempt is made to include this article, so that the review provides an up-to-date and comprehensive picture, unless the authors are satisfied that this article provides no new insights.	Based on your feedback and that of the other reviewers we have now included the following points in the limitations section of the paper:  • The potential absence of studies, which may have been included in the grey literature which we omitted: "We did not conduct an exhaustive search for grey literature sources due to the challenges in appraising these types of publications as well as the lack of standardised search guidelines for scoping reviews (Tricco et al 2016)." (Page 22)

		 • Limitations of the included studies, such as the lack of reporting on certain issues: “Finally, given the nature of scoping reviews, a critical appraisal of the studies included in the review was not performed.²⁶ This could be perceived as a limitation since the overall quality and level of detail of the studies was variable. There was also significant heterogeneity between studies, which makes direct comparisons difficult. Future work should aim to clearly outline the context in which CHWs work and provide a detailed description of their job roles and responsibilities to help orientate the reader and contextualise the setting.” (Page 22) Although we were originally unable to find someone to reliably translate the paper by Sylla et al., from French to English we have now done this and included this study within the final review as you requested (See Table 3, Supplementary Material)
5	As hinted to by the authors, the term CHW is a catch all for a very diverse category of informal health workers. I suspect the authors are au fait with this as the search terms used (synonyms for CHWs) is quite extensive and covers many of the key ones. I think it would be useful to write a bit more in the introductory section to elaborate this diversity particularly for the non-specialist reader.	Thank you for this helpful comment. We have now expanded on this point in the introduction section to ensure clarity for non-specialist readers: “CHW is an umbrella term for lay people working within their own community in a health promotion, prevention and delivery role,³ however the nomenclature used to describe CHWs is wide ranging and their exact roles, responsibilities, recruitment, remuneration and training vary from country to country” (Page 5) We have also included the following reference to the 2017 paper by Olaniran et al, “Who is a community health worker? – a systematic review of definitions” for readers who are interested to learn more

		about CHW nomenclature, roles and responsibilities. (Full reference: Olaniran, A., Smith, H., Unkels, R., Bar-Zeev, S., & van den Broek, N. (2017). Who is a community health worker? - a systematic review of definitions. Glob Health Action, 10(1), 1272223. doi: 10.1080/16549716.2017.1272223) We have also added two columns to Table 3 in the supplementary material outlining the terms used to describe CHWs in each individual study and a description of the cadre.
6	The other significant limitation is the exclusion of grey literature. In the development context, there is considerably more published in NGO / IGO repositories, much of which has direct relevance to practitioners and local policymakers. I think this nuance needs to be highlighted.	Admittedly we may have missed some literature since our search of the 'grey literature' was limited to the following databases and search engines:  • E-theses online service (ETHoS); • Conference proceedings on Index of Conference proceedings; • Google Scholar. To help clarify this for the reader we have added a section to the limitations section of the paper stating: “We did not conduct an exhaustive search for grey literature sources due to the challenges in appraising these types of publications as well as the lack of standardised search guidelines for scoping reviews (Tricco et al 2016).” (Page 22) Although we may have missed out reports that may have been indexed in NGO / IGO repositories it is likely that they would have not been included since they would have fallen under the category of a report rather

		than a research study (see exclusion criteria and Minor Changes Comment number 6 from Reviewer 3).
7	In the discussion section, the authors make a point that NCD is a priority and that maternal and child health, and HIV/TB seem to predominate. I agree. That said I think it would be useful to illustrate the context better - CHWs exist and are often focused on certain disease programmatic areas because they are cheaper to train and roll out than nursing, medical or paramedical staff in LMICs. The public health need now to reorientate CHW areas of work towards NCDs is essential but it also challenging as it will require retraining and refocusing large CHW workforces. This challenge is not well made in the discussion.	We agree with your comment that training the workforce to deal with NCDs is vitally important, although challenging, and have integrated this important point into the discussion section by stating: “Given the combined shortage of a lack of specialist health workers and the high morbidity and mortality from the aforementioned disease groups, CHWs have rightly been trained to address these issues. Although the burden of infectious disease and child and maternal health remain problematic in LMIC settings...” and “This public health need to expand CHW provision towards NCDs is both an opportunity and challenge, since it will require the commitment of government, funders and program managers to retrain and refocus large CHW workforces.” (Page 17) We hope the additions and changes we have made as outlined above now help to clarify your points and thank you for making this useful addition to our work.
8	There was also discussion around the absence of learning theories to underpin the on-going training. That said, the absence of evidence does not necessarily equate to evidence of absence. Neither can it be said that the training approaches adopted are inappropriate or ineffectual on the basis of whether there is an underpinning learning theory described? I don't think	Given that we have now redefined the aims of our work and reanalysed the included materials we have made the decision to remove the discussion around learning theories from the manuscript. Although we initially felt this was an important point to discuss, we agree that the lack of evidence means that a strong conclusion cannot be drawn as to the merits or disadvantages for the use of

	on the basis of what evidence is given in the results section a strong assertion can be made on this. In a similar vein, the authors hint that 'protocol-based' training is inferior or somehow inappropriate. In an ideal situation perhaps. That said, as the authors argue context is everything: do protocol base training approaches predominate because the target audience has low literacy or understanding of science, or there is a higher policy goal of quality assurance and governance that is driving the want for standardized CHW responses rather than autonomy of treatment choices?	learning theories. Regarding your second point as to why protocol based approaches seem to be commonplace in the literature, this is a very interesting point. We feel that this could probably a combination of the two reasons you mentioned as to why this takes place, however, without clear evidence around this it is difficult for us to objectively comment. This is an area of open research to which we would like to contribute in future work. Many thanks for the insightful point here.
9	The findings and conclusions suggest little robust work in this field but also difficulties making comparisons due to the heterogeneity of the articles. Perhaps the authors may want to consider advocating the value of a realist approach in future research in this field as that may make better sense of the varied data, in terms of the differing contexts, generative mechanisms and their relation to outcomes?	Thank you for this extremely useful suggestion. We have re-written the discussion and conclusions section of the manuscript and as such have added this useful point into the conclusion. Given the heterogeneity of the field we advocate for a realist approach to evaluation for future research, considering training as a complex intervention. This may help those interested in the field to make better sense of its complex nature with view to understanding what works, for whom, and under what conditions. Through taken this approach and considering the contextual requirements, on-going training programmes are more likely to contribute to a systems level improvement in resource limited settings. (Page 23)

Reviewer: 2

Reviewer Name: Joseph Mumba Zulu

Institution and Country: University of Zambia, School of Public Health, Zambia

Competing Interests: N/A

Reviewer 2 Comment Number	Reviewer Comment	Author response
1	This is an important review as it focuses on contributing towards addressing the knowledge gap on refresher trainings among community health workers. This information is a vital considering the increased call or emphasis on the need to promote community-based health systems and workforce.	Thank you for acknowledging the importance of this review. Given the growing emphasis being placed on the role of CHWs to deliver healthcare we certainly feel our work will be a useful addition to the literature. We also wish to thank you for your insightful comments. All changes to the manuscript have been made using the 'track changes' feature in Microsoft Word and changes to the supplementary material are highlighted in red. To assist you with your review we have responded to each of your comments in turn within this table, with cross-references to the main text or supplementary material where appropriate.
2	There is need to restructure and strengthen the results section in line with the objective of the review. For now, much information has been placed in the discussion section compared to the results section. There is need to have subheadings in the results section in line with the scoping review objectives or questions, which are: How is on-going training for CHWs	Thank you for this point. Given the feedback by review one and review three, we have changed the aims of the paper to state: "The aim of this systematic scoping

	designed and delivered? Are theories of learning used to inform the design of on-going training? Do mobile technologies have a role in the delivery of on-going training? How are the outcomes of on-going training reported?	review was therefore to map the current delivery, implementation and evaluation of on-going training provision for CHWs in LMICs.” (Page 6) As such we no longer have pre-defined themes to explore, however we have outlined the resulting themes in the discussion section by including subheadings as you have suggested. We have also included a comprehensive summary table in the supplementary material which outlines the features you mention such as how on-going training for CHWs is delivered, whether or not theories of learning are used to inform the design of on-going training, whether mobile technologies are used and how the outcomes of on-going training are reported (see supplementary material, Table 3).
3	In addition, to these questions, the authors may consider including information or a paragraph on the key topics that the community health workers were re-trained in and where possible the duration of the training. This information could be part of the heading under design and delivery of CHW re-training.	These details are included within Table 3 in the supplementary text. The details include disease focus area, type of on-going training, duration of on-going training and outcome measures of assessment. Since these details required a large word count total we felt they were best presented in a uniform manner in a table within the supplementary material. We hope that the reader will be able to easily cross-reference studies using this table. (See supplementary material, Table 3).

4	Mention/state the total number of articles that were included for review in abstract.	The abstract reads as: “The scoping review found 35 original studies that met the inclusion criteria.” (Page 2) We hope this clarifies your point.
5	Search terms: while the authors state in the results section that about 17 different terms were identified for community health workers during the search process, the authors do not mention these terms. It will be important for the authors to outline these terms in the methods section – and list them under the search terms used (since the community health worker concept is inadequate as a search term for this kind of study)	We included all of the search terms for ‘Community Health Worker’ in the supplementary material included in the original submission. We stated in the methods section “37 relevant search terms for ‘Community Health Workers’ and ‘on-going training’ were developed (see Table 1 in the supplementary material for the full list of terms used within the search strategies).” (Page 8) In terms of results we did indeed identify 17 different terms used by the authors of the included studies to describe CHWs. As such we have included these in a table within the supplementary material, as well as a description of the CHW cadre (see supplementary material, Table 3).
6	Please start the duration for the scoping review	This is stated in the methods section of the paper under the subheading “Search Strategy and Selection criteria”: “The following databases were searched to identify primary, peer-

		reviewed studies published from 12th September 1978, up to and including July 10th 2017". (Page 8)
7	There is a statement that the scoping review is advantageous because of its "less emphasis on the critical appraisal of the included evidence compared to a traditional systematic review", it would nice if the authors could provide more information on the advantages of this design?	Thank you for this important point which is relevant since scoping reviews have only recently started to become a popular means for synthesizing and presenting data. We have now referenced the 2010 paper by Levac et al., which outlines some of the key advantage of a scoping review – namely that scoping reviews are beneficial in mapping the "extent, range, and nature of research activity" in a field with emerging evidence. (Page 7) Given the relative lack of evidence surrounding on-going training for CHWs within LMICs and no previous review in this field we felt a scoping review was appropriate for this work. Additional reference Levac, D., Colquhoun, H., & O'Brien, K. K. (2010). Scoping studies: advancing the methodology. Implement Sci, 5, 69. doi: 10.1186/1748-5908-5-69
8	I recommend moving the PRISMA FLOW CHART or steps taken in selecting the final articles from the results section to the methods section.	We have referred to the PRISMA statement guidelines, which advise that the PRISMA flow chart depicting study selection should be included within the results section of the paper (http://www.prisma-statement.org/documents/PRISMA%20EandE%202009.pdf).

9	There is need to add data analysis section.	Thank you for this important point which Reviewer 3 also made. We have now added a data analysis sub-section in the Methodology section of the paper. This reads as follows: Data analysis Once studies were determined to have met the inclusion criteria, the relevant data was systematically extracted from each study and tabulated using a 'data charting form' in a Microsoft Excel spreadsheet by one author (JOD). The data extracted from each study included the study author, title, date, country and region which the study took place, CHW name and cadre description, the number of CHWs who took part in the study, the disease focus area, a description of how the on-going training programme was delivered, as well as a report on the outcomes measured. The use of a 'data charting form' has been recommended by Arksey & O'Malley and Levac et al., as a key stage of conducting a scoping review.^{26,32} Where necessary, the corresponding authors for relevant studies were contacted via email to clarify aspects of their work prior to final inclusion. Once the data had been transferred into the spreadsheet, two authors (JOD & NW) reviewed the information and selected key focus areas for the review, as well as categories for the outcome reporting methods. The same two authors

		thematically grouped outcome data from on-going training into one of the following four categories: 1. Knowledge and Skills Assessments 2. Changes in Behaviour, Attitudes or Practice 3. Qualitative Assessments 4. Mixed Methods Approaches. Similarly, if the use of mobile technologies was noted in the study, this was documented and categorised using the mHealth framework developed by Labrique et al.³³ (Page 12-13) Additional References  - Arksey, H., & O'Malley, L. (2005). Scoping studies: towards a methodological framework. International Journal of Social Research Methodology, 8(1), 19-32. doi: 10.1080/1364557032000119616 - Levac, D., Colquhoun, H., & O'Brien, K. K. (2010). Scoping studies: advancing the methodology. Implement Sci, 5, 69. doi: 10.1186/1748-5908-5-69 - Labrique, A. B., Vasudevan, L., Kochi, E., Fabricant, R., & Mehl, G. (2013). mHealth innovations as health system strengthening tools: 12 common applications and a visual framework. Glob Health Sci Pract, 1(2), 160-171. doi: 10.9745/ghsp-d-13-00031
--	--	---

--	--	--

Reviewer: 3

Reviewer Name: Brynne Gilmore

Institution and Country: Trinity College Dublin, Ireland

Competing Interests: I frequently co-author and am currently involved in several research projects with a researcher (Dr. Vallieres) that was first author on one of the included studies. No other conflicts of interest.

Reviewer 3 Comment Number	Reviewer Comment	Author response
1	Thank you for the opportunity to read this very interesting study and manuscript. As the authors highlighted, training of CHWs is a very important issue that this manuscript works to address. I hope that you find the below comments helpful in your revisions.	Thank you again for taking the time to review our work Dr. Gilmore. We are really appreciative of your thorough comments, which have undoubtedly helped to strengthen our work. Please find below the response to your comments. All changes to the manuscript have been made using the 'track changes' feature in Microsoft Word and changes to the supplementary material are highlighted in red.
2	References are mixed up and very confusing. The included articles seem to be referenced separately in the supplementary files from those that are in-text citations, but this makes for duplication. For example, when citing included study in text it might be 31, then in the supplementary file it is 10, so I cannot refer to the data table to get more information. Please fix.	Apologies for this confusion and thank you for the suggestion. We have now amended this error to ensure the references in the main file correspond to those in the supplementary material. We have made the decision to remove the reference list from the supplementary file and just refer to the single reference list in the main text. We hope this will make it easier for the reader to cross-reference between the main text and the material contained

		within supplementary file. (See Table 3, Supplementary Material)
3	Pg. 6 – the research questions seem a bit repetitive (1 and 2). Suggest wording as they were in the abstract. Additionally, question 2 the “Are participatory approaches used to engage communities to ensure context-relevant design” – is new (not in abstract), and also does not necessarily fit with theories. Programmes can use theories, without engaging communities. Should this be a separate question, especially as you title them separately in the Discussion?	We also received feedback from Reviewer One (see Reviewer One, Comment 2) that our initial research questions were somewhat confusing. Given this feedback we decided to rework the initial research questions, and refocus the aim of the paper to read as follows: “The aim of this systematic scoping review was therefore to map the current delivery, implementation and evaluation of on-going training provision for CHWs in LMICs.” (Page 6) This is also reflected in the amended abstract (Page 2). We hope this will clarify the aim of the paper for the reader. The resulting themes, such as the type of on-going training, delivery of on-going training and the use of mobile technologies etc. are detailed within the results and discussion section of the paper as key points that came out of the review.
4	Much more information is needed on the data extraction and analysis of the included studies. How was the data from included studies systematically extracted? More details on this is needed (i.e. was a	Thank you for highlighting this omission. Reviewer 2 also requested that we added more information to regarding data analysis and so we

	template used, and how was this developed, how many people extracted etc). Once extracted, how was this analysed?	have added a sub-section within the 'Methodologies' section entitled 'Data Analysis' which reads as follows: Data analysis Once studies were determined to have met the inclusion criteria, the relevant data was systematically extracted from each study and tabulated using a 'data charting form' in a Microsoft Excel spreadsheet by one author (JOD). The data extracted from each study included the study author, title, date, country and region which the study took place, CHW name and cadre description, the number of CHWs who took part in the study, the disease focus area, a description of how the on-going training programme was delivered, as well as a report on the outcomes measured. The use of a 'data charting form' has been recommended by Arksey & O'Malley and Levac et al., as a key stage of conducting a scoping review.^{26,32} Where necessary, the corresponding authors for relevant studies were contacted via email to clarify aspects of their work prior to final inclusion. Once the data had been transferred into the spreadsheet, two authors (JOD & NW) reviewed the information and selected key focus areas for the review, as well as categories for the outcome reporting methods. The same two authors thematically grouped outcome data from on-going training into one of the following four categories: 1. Knowledge and Skills Assessments 2. Changes in Behaviour, Attitudes or Practice 3. Qualitative Assessments 4. Mixed Methods Approaches. Similarly, if the
--	--	--

		use of mobile technologies was noted in the study, this was documented and categorised using the mHealth framework developed by Labrique et al.³³ (Page 12-13) Additional References - Arksey, H., & O'Malley, L. (2005). Scoping studies: towards a methodological framework. International Journal of Social Research Methodology, 8(1), 19-32. doi: 10.1080/1364557032000119616 - Levac, D., Colquhoun, H., & O'Brien, K. K. (2010). Scoping studies: advancing the methodology. Implement Sci, 5, 69. doi: 10.1186/1748-5908-5-69 - Labrique, A. B., Vasudevan, L., Kochi, E., Fabricant, R., & Mehl, G. (2013). mHealth innovations as health system strengthening tools: 12 common applications and a visual framework. Glob Health Sci Pract, 1(2), 160-171. doi: 10.9745/ghsp-d-13-00031
5	The Results and Discussion are lacking. The results do not specifically answer your research question, it would be beneficial to draw back to the original questions.	Given that we have revised the aim of the paper we have now adjusted and enhanced the results and discussion section of the paper to reflect this. The resultant themes that emerged following the review have been detailed in the results section of the paper and have been discussed in more detail in the discussion (Pages

		17-22). We have made an effort to make more explicit reference to the studies included within the review. We have discussed the need for a greater use of qualitative research to enhance the current state of the field and we have made reference to Greenhalgh et al's., paper on this. We have also referenced your 2016 study where you call for a "shift from more traditional empirical studies to ones that consider the complex nature of such interventions and the importance of whole systems thinking. (Pages 19 and 20) Additional references - Gilmore, B., Adams, B. J., Bartoloni, A., Alhaydar, B., McAuliffe, E., Raven, J., . . . Vallieres, F. (2016). Improving the performance of community health workers in humanitarian emergencies: a realist evaluation protocol for the PIECES programme. BMJ Open, 6(8), e011753. doi: 10.1136/bmjopen-2016-011753 - Greenhalgh, T., Annandale, E., Ashcroft, R., Barlow, J., Black, N., Bleakley, A., . . . Ziebland, S. (2016). An open letter to The BMJ editors on qualitative research. BMJ, 352, i563. doi: 10.1136/bmj.i563
6	The results at times are also just superficial reporting – there is not a lot of information given with them. Even in the Supplementary File 2, the reader does not get enough information to really understand the training. I would suggest giving more details to more fully engage with the included	Thank you for asking us to expand upon our results. We have gone back to each individual study and included many more details in Table 3 within the supplementary materials regarding the content, duration, location and provider responsible for delivering on-

	studies and the topic of on-going training.	going training as well as the reported outcomes from each study (see supplementary material, Table 3) We have also reworked the discussion section of the paper to reflect the resulting themes that emerged from this analysis (see Discussion section of the main manuscript). We have added additional details about the negative aspects of on-going training that were reported so as to balance the argument and highlight key areas that should be considered.
7	Additionally, a lot of your discussion could belong to results, as not a lot of new literature is brought in to discuss results, it is more of a second reporting of the findings. For instance, Pg 16 paragraph starting on line 45 could all be moved to results. As well as pg. 17 paragraph from line 13 to 32.	Based on your feedback and reviewer 2's feedback (see comment number 2) we have restructured the results section of the paper, moving much of what was previously in the discussion into the results section. One example of this was where we modified and moved the details included originally in the discussion on page 16 line 45 and page 17 line 13-32 to the results section as you have requested. (See Results section of the main manuscript)
8	I think a description of the CHW cadre within the programme should be brought into the Table to provide more context.	Thank you for making this suggestion. We have attempted to do this by going back to each individual study to identify whether or not the description of the CHW cadre was mentioned. Where possible we have now included details on:  - Selection criteria - Roles and responsibilities - Whether or not CHW were paid - The number of households/people CHW were responsible for

		 - Level of education - Demographic details - Level and content of pre-service training Although we have attempted to do this for every study included within the final analysis, the information was not available for all of these domains in many of the studies. In some cases no description of the CHW cadre within the programme was provided. If this was the case we have documented it as “No details provided”. Additionally we have recognised this within the limitations section of the paper. Please refer to the column labelled ‘Cadre Description’ in Table 3 which is contained within the supplementary material. Based on this exercise we had to exclude two studies. Given our description of CHWs as having received “no formal professional or paraprofessional certificated or degreed tertiary education,” we had to exclude the studies by Gill and Sabin since the ‘Community Based Physicians Associates (CBPAs)’ were professionals who had received a ‘two year graduated from an accredited 2-year medical training program’ (We have clarified this in the exclusion criteria – page 9)
9	Also, articles that did bring in theories of learning and/or participatory approaches and/or mobile technology, should be indicated in the table. Right now all the pieces are somewhere in the article, but this table should bring them all together.	Thank you for this suggestion. We have now added an additional column to Table 3 in the supplementary material to indicate whether mobile technologies were referenced. (Please refer to supplementary material, Table 3) We have removed the column regarding the use of learning theories due to the feedback from Reviewer 1

		(see Reviewer 1, Comment 8)
10	The actual on-going training description is missing. I would hope that one could turn to your article to get details on what training was occurring, but as of now it is quite limited. More can also be brought in on the type of on-going (i.e. who conducted it? What specifically was it on (i.e. Some refresher training focus on 1 unit of a CHW package, others all skills). Who were the supervisors? Where did the supervision and/or training occur?)	Thank you. As per your suggestion We have now amended the training column within Table 3 to give it a more uniform structure and included sub-headings within each cell to outline (i) the type of on-going training; (ii) the context of training (iii) the duration of training; (iv) the individual or organization responsible for training delivery; (v) location of training. (Table 3, Supplementary Material)
11	The 'Outcomes' question needs some more thought I think. There is no clear link between the outcomes and the on-going training, just reporting of what was looked at. While I understand it is a scoping review, so you are reporting what is happening, if an article found no impact on outcomes of a training, I feel like this would be important to note within the article.	To help clarify this we have added a column in the supplementary materials table entitled 'Outcome measures and outcomes'. Here we outline the outcome measures that were used to measure on-going training, as well as the impact of on-going training, including where there was no impact reported (see Table 3, supplementary material column on outcomes and outcome reporting) We have also added additional details into the discussion section of the paper detailing outcomes, including any negative reports of on-going training.
12	Please double-check all of your data extraction – for the Vallieres article, the 'Outcome' you note is "Qualitative Assessment. Self reported measures of work engagement and job satisfaction". This was done through a self-reported survey – which is quantitative. If you read on, it is all very quantitative with scale validation and statistical reporting.	Thank you for highlighting this. We apologise for this mistake and we fully agree that the use of the Volunteer Functions Inventory, Perceived Supportive Supervision Scale, the Utrecht Work Engagement Scale and the Minnesota Satisfaction Questionnaires were all quantitative in

		nature. We have corrected this within the table to ensure it is an accurate reflection of the outcome measure, as well as double-checked all other included articles. We have rephrased the outcome measure of 'Change in behaviour and practice' to 'Change in behaviours, attitudes or practice'. We have classified studies which used a quantitative approach to work engagement and satisfaction, for example through surveys and questionnaires, under this heading. Studies which used a purely qualitative approach e.g. in-depth interviews and resulting thematic analysis, remained under the outcome heading 'Qualitative Assessment'. There were two other errors in the original manuscript that have now been modified. The Kawasaki study was originally reported as a qualitative assessment, however it was a combined qualitative and behaviour/practice change assessment. This has now been amended accordingly within the text. Similarly the Mash study was originally reported as using a knowledge and skills assessment, whereas in reality it used both a knowledge and skills assessment combined with qualitative feedback from supervisors. (See Table 3, Supplementary Material)
13	Conclusion – second paragraph. A lot of these ideas (partnerships, interdisciplinary approach, contextual needs, systems level improvement etc), are newly introduced. As well as the recommendations for research. These belong more in the discussion, and	Thank you for suggesting this revision. We have now moved the recommendations for future research to the discussion section of the paper, as well as rewritten the conclusion so

	then can be reiterated in the conclusion.	as not to introduce any new points that were not previously raised. (Page 22)
Minor changes		
1	Pg. 2. Switching between low-or middle-income and low-and middle-income (i.e. pg. 2, pg.3). Please pick one and change throughout for consistency.	We have altered this and for consistency with our key words changed this instance to 'low- and middle-income'. (Page 2)
2	Pg. 5 – CHWs often do more than health promotion and prevention – i.e. treatment, testing, surveillance... This is true of the CHWs in some of your included papers.	Thank you for clarifying this. We have now amended this sentence to read: “In the broadest sense, CHW is an umbrella term for lay people working within their own community in a health promotion, prevention and delivery role, however the nomenclature used to describe CHWs in wide ranging and their exact roles, responsibilities, recruitment, remuneration and training vary from country to country” (Page 5) By including the term 'delivery' alongside promotion and prevention we feel this encompasses some of the other key roles CHWs play which you mentioned. We have also caveated this sentence by stating that their exact responsibilities vary.
3	Pg. 5 – implying that CHWs are all voluntary (line 13-17, and then comparing to 'salaried health workers' in line 23). This is not true of all CHWs – and some of your included studies have salaried CHWs (i.e. HEWs in Ethiopia). Please clarify.	The aim of this scoping review was to provide a broad overview of on-going training regardless of whether or not CHWs were salaried and we have now clarified this by removing the reference to voluntary CHWs. We felt it was important to include both paid and unpaid cadres of CHWs in this scoping review since many LMICs are moving towards a mixed

		model of salaried and volunteer CHWs (see - Kok, M. C., Broerse, J. E. W., Theobald, S., Ormel, H., Dieleman, M., & Taegtmeier, M. (2017). Performance of community health workers: situating their intermediary position within complex adaptive health systems. Hum Resour Health, 15(1), 59. doi: 10.1186/s12960-017-0234-z) We have also removing the statement regarding 'cost-effectiveness in comparison to other salaried health workers' since we have included non-salaried cadres of CHWs within this scoping review. It now reads: “When provided with the correct resources, training, and support, CHWs have been proven to help improve health outcomes and accessibility to basic services.^{2,6,7”} (Page 5)
4	There are lots of quotes within the paper, but none of page numbers of the reference. Please make sure this is OK with BMJ guidelines, and if not add page numbers to all in-text quotes.	We have checked the BMJ Open referencing guidelines which state the Vancouver style should be used and “the authors' names are followed by the title of the article; the title of the journal abbreviated according to the style of Index Medicus; the year of publication; the volume number; and the first and last page numbers.” As such we have ensured that the first and last page numbers accompany each reference. There are no specific guidelines from BMJ regarding individual page numbers for in-text

		quotations.
5	Please consider reworking you Supplementary material. Right now it is a little difficult to follow. Also, within reporting supplementary material, please keep the order in-line with the methodology and when it is reported in the article (i.e. move the search terms before the search results).	Thank you for this helpful suggestion. We have changed how the supplementary material has been presented and followed your suggestion to move the search terms before the search results. Furthermore we have included all search strategies within one table to make this easier for the reader to navigate (Table 1, Supplementary Material).
6	Your description of 'grey literature' is not really consistent with what most would think. Especially given your topic area, grey literature would often include NGO reports, policy documents, programme descriptions and evaluations etc. But these were 1) not searched for looking at your grey lit sources and 2) would have been excluded based on exclusion #3 and research design. So I don't think you include 'grey' literature, but included non-peer reviewed research. At very least, I think you need to specify what type of grey literature.	Although informal reports and policy documents may have contained some useful information regarding on-going training, as you have pointed out these would not have been included due to the exclusion criteria #3. Given that we did not search grey literature sources such as NGO/IGO repositories, we have clarified this within the limitations section of the paper by stating: "We did not conduct an exhaustive search for grey literature sources due to the challenges in appraising these types of publications as well as the lack of standardised search guidelines for scoping reviews (Tricco et al 2016)." (Page 22) We have also rephrased our search of the 'grey literature' to the 'additional non-peer reviewed literature', so as to try and avoid any confusion.
7	Pg. 14, line 28 – it is not Table 1, as there is a Table before this in supplementary. Again, please consider	Since reworking the supplementary material we have renumbered the tables, as well as adjusting this within

	reworking the supplementary file for more clarity.	the text of the main manuscript. Please see our response to 'Minor revisions point 5' for more details regarding how we have revised the supplementary material.
8	Mobile training in results – what kind of mobile training was used?	We have tried to clarify this both within the results section of the paper and also by adding an extra column to the supplementary material Table 3 which provides a broad overview of what mobile training was used. We have categorised the use of mobile training using the Labrique framework for mHealth tools and included the categories within the supplementary material in Table 3. (Reference - Labrique, A. B., Vasudevan, L., Kochi, E., Fabricant, R., & Mehl, G. (2013). mHealth innovations as health system strengthening tools: 12 common applications and a visual framework. Glob Health Sci Pract, 1(2), 160-171. doi: 10.9745/ghsp-d-13-00031)
9	Figure 1 – what are the 'other sources' is that the grey searching plus snowballing?	Yes – this was the 'grey literature' plus snowballing. We have now clarified this in the revised version of the PRISMA diagram to a search of the 'additional non-peer reviewed literature plus snowballing' (please see PRISMA diagram)
10	Supplementary material – search results and strategy. Please do not combine databases when reporting the number of hits. Numbers for each separate search should be reported.	We have repeated the search strategy for each database individually and reported the individual results (see Supplementary Material Table 2)

11	If your search is sensitive, I am cautious that BEI only have 1 hit? (and ERIC and LILACS seem low too). Do you have any explanation for this?	We have repeated the search on BEI, ERIC and LILACS and the number of hits were incorrectly reported in the initial submission for BEI (1) and ERIC (9). The correct number of hits has for BEI (38) and ERIC (262) now been updated both within the text and the PRISMA diagram. We apologise for this error and as such we have gone back and re-searched all of the databases to ensure the number of reported hits is correct. The remaining databases were reported correctly, including LILACS. Once the results from BEI and ERIC were de-duplicated and searched no new original studies which met the inclusion criteria were found that had not already been included in the review.
	Page 24, line 14 – “settings of poverty and inequality”. I would change this. It was never brought up before, only that your setting was LMIC. Thus, it reads as you are stating all LMICs are settings of poverty and inequality. What about HICs, there are not settings of poverty and inequality within these? So if this is what you are looking at then these settings should be included too.	Apologies for this inclusion. We fully accept this point and as such have removed this comment from our paper. It now reads as “This fragmented approach means little is understood about how to best deliver on-going training in LMICs.” (Page 23)
Additional comments		
1	Pg. 8 – I would reorder the paragraph at line 42, moving the first sentence to after the second sentence. ... “....are not applied to scoping reviews. Nonetheless, our review followed...., as demonstrated in the following sections.”	Thank you for this suggestion. We have enacted this change. The text now reads:

		“A review protocol was not published, and the study was not registered with PROSPERO, as these mechanisms are not applied to scoping reviews.^{25,26} Nonetheless, our scoping review followed explicit and transparent research steps to explore the research evidence on on-going training for CHWs in LMICs, as demonstrated in the following sections.’(Page 7)
2	Pg 13 – the ‘third reviewer’ was independent, as in not part of the research team? Was it always the same person, or how did you find them?	Yes, the third review was always the same person. They were not part of the direct research team who carried out this study. Rather, they were a medical doctor with research background whom the lead author has previously collaborated with. We have now clarified this within the text, by stating: “The independent researcher was always the same person and was not part of the direct research team listed in this study.” (Page 12)
3	Pg. 14, line 13 – Sentence “studies were initially....” Is repetitive with methods section. Results should be reporting only, not describing the screening process again.	Thank you for mentioning this. We have removed this section describing the screening process from the results section. The text now reads: “After exclusion of duplicate references using the EndNote referencing system, 2609 papers were identified for initial screening. After the initial abstract and title screen, 172 studies were identified for full text review. Following this review, 137 papers were excluded.” (Page 14)

4	Pg. 14, line 17 – consider rephrasing “in-depth analysis”, as it could seem that at this point data was analysed, as opposed to articles reviewed/examined.	We have corrected this wording to ‘review’, rather than ‘in-depth analysis’. “Following this review, 137 papers were excluded.” (Page 14)
---	---	--

VERSION 2 – REVIEW

REVIEWER	Brynne Gilmore Trinity College Dublin, Ireland
REVIEW RETURNED	12-Mar-2018

GENERAL COMMENTS	Dear Dr. O’Donovan and authors: Thank you for the detailed response noting the changes made. I think you did a great job addressing all reviewers comments and I am very satisfied with your responses/changes to my comments. Good luck with the rest of the submission!
---

REVIEWER	Joseph M Zulu University of Zambia, Zambia
REVIEW RETURNED	14-Mar-2018

GENERAL COMMENTS	I have no further comments on the manuscript
--